# Learning with Pseudo-Ensembles

**Philip Bachman**
McGill University
Montreal, QC, Canada
phil.bachman@gmail.com

**Ouais Alsharif**
McGill University
Montreal, QC, Canada
ouais.alsharif@gmail.com

**Doina Precup**
McGill University
Montreal, QC, Canada
dprecup@cs.mcgill.ca

## Abstract

We formalize the notion of a *pseudo-ensemble*, a (possibly infinite) collection of *child models* spawned from a *parent model* by perturbing it according to some *noise process*. E.g., dropout [9] in a deep neural network trains a pseudo-ensemble of child subnetworks generated by randomly masking nodes in the parent network. We examine the relationship of pseudo-ensembles, which involve perturbation in model-space, to standard ensemble methods and existing notions of robustness, which focus on perturbation in observation-space. We present a novel regularizer based on making the behavior of a pseudo-ensemble robust with respect to the noise process generating it. In the fully-supervised setting, our regularizer matches the performance of dropout. But, unlike dropout, our regularizer naturally extends to the semi-supervised setting, where it produces state-of-the-art results. We provide a case study in which we transform the Recursive Neural Tensor Network of [19] into a pseudo-ensemble, which significantly improves its performance on a real-world sentiment analysis benchmark.

## 1  Introduction

Ensembles of models have long been used as a way to obtain robust performance in the presence of noise. Ensembles typically work by training several classifiers on perturbed input distributions, e.g. bagging randomly elides parts of the distribution for each trained model and boosting re-weights the distribution before training and adding each model to the ensemble. In the last few years, dropout methods have achieved great empirical success in training deep models, by leveraging a noise process that perturbs the model structure itself. However, there has not yet been much analysis relating this approach to classic ensemble methods or other approaches to learning robust models.

In this paper, we formalize the notion of a *pseudo-ensemble*, which is a collection of child models spawned from a parent model by perturbing it with some noise process. Sec. 2 defines pseudo-ensembles, after which Sec. 3 discusses the relationships between pseudo-ensembles and standard ensemble methods, as well as existing notions of robustness. Once the pseudo-ensemble framework is defined, it can be leveraged to create new algorithms. In Sec. 4, we develop a novel regularizer that minimizes variation in the output of a model when it is subject to noise on its inputs and its internal state (or structure). We also discuss the relationship of this regularizer to standard dropout methods. In Sec. 5 we show that our regularizer can reproduce the performance of dropout in a fully-supervised setting, while also naturally extending to the semi-supervised setting, where it produces state-of-the-art performance on some real-world datasets. Sec. 6 presents a case study in which we extend the Recursive Neural Tensor Network from [19] by converting it into a pseudo-ensemble. We

generate the pseudo-ensemble using a noise process based on Gaussian parameter fuzzing and latent subspace sampling, and empirically show that both types of perturbation contribute to significant performance improvements beyond that of the original model. We conclude in Sec. 7.

## 2    What is a pseudo-ensemble?

Consider a data distribution $p_{xy}$ which we want to approximate using a parametric *parent model* $f_\theta$. A pseudo-ensemble is a collection of $\xi$-perturbed *child models* $f_\theta(x; \xi)$, where $\xi$ comes from a *noise process* $p_\xi$. Dropout [9] provides the clearest existing example of a pseudo-ensemble. Dropout samples subnetworks from a source network by randomly masking the activity of subsets of its input/hidden layer nodes. The parameters shared by the subnetworks, through their common source network, are learned to minimize the expected loss of the individual subnetworks. In pseudo-ensemble terms, the source network is the *parent model*, each sampled subnetwork is a *child model*, and the *noise process* consists of sampling a node mask and using it to extract a subnetwork.

The noise process used to generate a pseudo-ensemble can take fairly arbitrary forms. The only requirement is that sampling a noise realization $\xi$, and then imposing it on the parent model $f_\theta$, be computationally tractable. This generality allows deriving a variety of pseudo-ensemble methods from existing models. For example, for a Gaussian Mixture Model, one could perturb the means of the mixture components with, e.g., Gaussian noise and their covariances with, e.g., Wishart noise.

The goal of learning with pseudo-ensembles is to produce models robust to perturbation. To formalize this, the general pseudo-ensemble objective for supervised learning can be written as follows[1]:

$$\operatorname*{minimize}_{\theta} \ \mathbb{E}_{(x,y) \sim p_{xy}} \ \mathbb{E}_{\xi \sim p_\xi} \ \mathcal{L}(f_\theta(x; \xi), y), \tag{1}$$

where $(x, y) \sim p_{xy}$ is an (observation, label) pair drawn from the data distribution, $\xi \sim p_\xi$ is a *noise realization*, $f_\theta(x; \xi)$ represents the output of a child model spawned from the parent model $f_\theta$ via $\xi$-perturbation, $y$ is the true label for $x$, and $\mathcal{L}(\hat{y}, y)$ is the loss for predicting $\hat{y}$ instead of $y$.

The generality of the pseudo-ensemble approach comes from broad freedom in describing the noise process $p_\xi$ and the mechanism by which $\xi$ perturbs the parent model $f_\theta$. Many useful methods could be developed by exploring novel noise processes for generating perturbations beyond the independent masking noise that has been considered for neural networks and the feature noise that has been considered in the context of linear models. For example, [17] develops a method for learning "ordered representations" by applying dropout/masking noise in a deep autoencoder while enforcing a particular "nested" structure among the random masking variables in $\xi$, and [2] relies heavily on random perturbations when training Generative Stochastic Networks.

## 3    Related work

Pseudo-ensembles are closely related to traditional ensemble methods as well as to methods for learning models robust to input uncertainty. By optimizing the expected loss of individual ensemble members' outputs, rather than the expected loss of the joint ensemble output, pseudo-ensembles differ from boosting, which iteratively augments an ensemble to minimize the loss of the joint output [8]. Meanwhile, the child models in a pseudo-ensemble share parameters and structure through their parent model, which will tend to correlate their behavior. This distinguishes pseudo-ensembles from traditional "independent member" ensemble methods, like bagging and random forests, which typically prefer diversity in the behavior of their members, as this provides bias and variance reduction when the outputs of their members are averaged [8]. In fact, the regularizers we introduce in Sec. 4 explicitly *minimize* diversity in the behavior of their pseudo-ensemble members.

The definition and use of pseudo-ensembles are strongly motivated by the intuition that models trained to be robust to noise should generalize better than models that are (overly) sensitive to small perturbations. Previous work on robust learning has overwhelmingly concentrated on perturbations affecting the inputs to a model. For example, the optimization community has produced a large body of theoretical and empirical work addressing "stochastic programming" [18] and "robust optimization" [4]. Stochastic programming seeks to produce a solution to a, e.g., linear program that performs

well *on average*, with respect to a known distribution over perturbations of parameters in the problem definition[2]. Robust optimization generally seeks to produce a solution to a, e.g., linear program with optimal *worst case* performance over a given set of possible perturbations of parameters in the problem definition. Several well-known machine learning methods have been shown equivalent to certain robust optimization problems. For example, [24] shows that using Lasso (i.e. $\ell_1$ regularization) in a linear regression model is equivalent to a robust optimization problem. [25] shows that learning a standard SVM (i.e. hinge loss with $\ell_2$ regularization in the corresponding RKHS) is also equivalent to a robust optimization problem. Supporting the notion that noise-robustness improves generalization, [25] prove many of the statistical guarantees that make SVMs so appealing directly from properties of their robust optimization equivalents, rather than using more complicated proofs involving, e.g., VC-dimension.

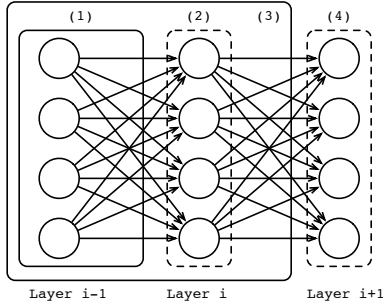

Figure 1: How to compute partial noisy output $f_\theta^i$: (1) compute $\xi$-perturbed output $\tilde{f}_\theta^{i-1}$ of layers $< i$, (2) compute $f_\theta^i$ from $\tilde{f}_\theta^{i-1}$, (3) $\xi$-perturb $f_\theta^i$ to get $\tilde{f}_\theta^i$, (4) repeat up through the layers $> i$.

More closely related to pseudo-ensembles are recent works that consider approaches to learning linear models with inputs perturbed by different sorts of noise. [5] shows how to efficiently learn a linear model that (globally) optimizes expected performance w.r.t. certain types of noise (e.g. Gaussian, zero-masking, Poisson) on its inputs, by marginalizing over the noise. Particularly relevant to our work is [21], which studies dropout (applied to linear models) closely, and shows how its effects are well-approximated by a Tikhonov (i.e. quadratic/ridge) regularization term that can be estimated from both labeled and unlabeled data. The authors of [21] leveraged this label-agnosticism to achieve state-of-the-art performance on several sentiment analysis tasks.

While all the work described above considers noise on the input-space, pseudo-ensembles involve noise in the model-space. This can actually be seen as a superset of input-space noise, as a model can always be extended with an initial "identity layer" that copies the noise-free input. Noise on the input-space can then be reproduced by noise on the initial layer, which is now part of the model-space.

## 4 The Pseudo-Ensemble Agreement regularizer

We now present Pseudo-Ensemble Agreement (PEA) regularization, which can be used in a fairly general class of computation graphs. For concreteness, we present it in the case of deep, layered neural networks. PEA regularization operates by controlling distributional properties of the random vectors $\{f_\theta^2(x;\xi), ..., f_\theta^d(x;\xi)\}$, where $f_\theta^i(x;\xi)$ gives the activities of the $i^{th}$ layer of $f_\theta$ in response to $x$ when layers $< i$ are perturbed by $\xi$ while layer $i$ is left unperturbed. Fig. 1 illustrates the construction of these random vectors. We will assume that layer $d$ is the output layer, i.e. $f_\theta^d(x)$ gives the output of the unperturbed parent model in response to $x$ and $f_\theta^d(x;\xi) = f_\theta(x;\xi)$ gives the response of the child model generated by $\xi$-perturbing $f_\theta$.

Given the random vectors $f_\theta^i(x;\xi)$, PEA regularization is defined as follows:

$$\mathcal{R}(f_\theta, p_x, p_\xi) = \mathop{\mathbb{E}}_{x \sim p_x} \mathop{\mathbb{E}}_{\xi \sim p_\xi} \left[ \sum_{i=2}^{d} \lambda_i \mathcal{V}_i(f_\theta^i(x), f_\theta^i(x;\xi)) \right], \tag{2}$$

where $f_\theta$ is the parent model to regularize, $x \sim p_x$ is an unlabeled observation, $\mathcal{V}_i(\cdot, \cdot)$ is the "variance" penalty imposed on the distribution of activities in the $i^{th}$ layer of the pseudo-ensemble spawned from $f_\theta$, and $\lambda_i$ controls the relative importance of $\mathcal{V}_i$. Note that for Eq. 2 to act on the "variance" of the $f_\theta^i(x;\xi)$, we should have $f_\theta^i(x) \approx \mathbb{E}_\xi f_\theta^i(x;\xi)$. This approximation holds reasonably well for many useful neural network architectures [1, 22]. In our experiments we actually compute the penalties $\mathcal{V}_i$ between independently-sampled pairs of child models. We consider several different measures of variance to penalize, which we will introduce as needed.

## 4.1 The effect of PEA regularization on feature co-adaptation

One of the original motivations for dropout was that it helps prevent "feature co-adaptation" [9]. That is, dropout encourages individual features (i.e. hidden node activities) to remain helpful, or at least not become harmful, when other features are removed from their local context. We provide some support for that claim by examining the following optimization objective [3]:

$$\underset{\theta}{\text{minimize}} \; \underset{(x,y)\sim p_{xy}}{\mathbb{E}} \left[\mathcal{L}(f_\theta(x), y)\right] + \underset{x\sim p_x}{\mathbb{E}} \underset{\xi\sim p_\xi}{\mathbb{E}} \left[\sum_{i=2}^{d} \lambda_i \mathcal{V}_i(f_\theta^i(x), f_\theta^i(x;\xi))\right], \tag{3}$$

in which the supervised loss $\mathcal{L}$ depends only on the parent model $f_\theta$ and the pseudo-ensemble only appears in the PEA regularization term. For simplicity, let $\lambda_i = 0$ for $i < d$, $\lambda_d = 1$, and $\mathcal{V}_d(v_1, v_2) = \mathcal{D}_{KL}(\text{softmax}(v_1) \| \text{softmax}(v_2))$, where softmax is the standard softmax and $\mathcal{D}_{KL}(p_1 \| p_2)$ is the KL-divergence between $p_1$ and $p_2$ (we indicate this penalty by $\mathcal{V}^k$). We use $\text{xent}(\text{softmax}(f_\theta(x)), y)$ for the loss $\mathcal{L}(f_\theta(x), y)$, where $\text{xent}(\hat{y}, y)$ is the cross-entropy between the predicted distribution $\hat{y}$ and the true distribution $y$. Eq. 3 never explicitly passes label information through a $\xi$-perturbed network, so $\xi$ only acts through its effects on the distribution of the parent model's predictions when subjected to $\xi$-perturbation. In this case, (3) trades off accuracy against feature co-adaptation, as measured by the degree to which the feature activity distribution at layer $i$ is affected by perturbation of the feature activity distributions for layers $< i$.

We test this regularizer empirically in Sec. 5.1. The observed ability of this regularizer to reproduce the performance benefits of standard dropout supports the notion that discouraging "co-adaptation" plays an important role in dropout's empirical success. Also, by acting strictly to make the output of the parent model more robust to $\xi$-perturbation, the performance of this regularizer rebuts the claim in [22] that noise-robustness plays only a minor role in the success of standard dropout.

## 4.2 Relating PEA regularization to standard dropout

The authors of [21] show that, assuming a noise process $\xi$ such that $\mathbb{E}_\xi[f(x;\xi)] = f(x)$, logistic regression under the influence of dropout optimizes the following objective:

$$\sum_{i=1}^{n} \underset{\xi}{\mathbb{E}} \left[\ell(f_\theta(x_i;\xi), y_i)\right] = \sum_{i=1}^{n} \ell(f_\theta(x_i), y_i)) + R(f_\theta), \tag{4}$$

where $f_\theta(x_i) = \theta x_i$, $\ell(f_\theta(x_i), y_i)$ is the logistic regression loss, and the regularization term is:

$$R(f_\theta) \equiv \sum_{i=1}^{n} \underset{\xi}{\mathbb{E}} \left[A(f_\theta(x_i;\xi)) - A(f_\theta(x_i))\right], \tag{5}$$

where $A(\cdot)$ indicates the log partition function for logistic regression.

Using only a KL-d penalty at the output layer, PEA-regularized logistic regression minimizes:

$$\sum_{i=1}^{n} \ell(f_\theta(x_i), y_i) + \underset{\xi}{\mathbb{E}} \left[\mathcal{D}_{KL} \left(\text{softmax}(f_\theta(x_i)) \| \text{softmax}(f_\theta(x_i;\xi))\right)\right]. \tag{6}$$

Defining distribution $p_\theta(x)$ as $\text{softmax}(f_\theta(x))$, we can re-write the PEA part of Eq. 6 to get:

$$\underset{\xi}{\mathbb{E}} \left[D_{KL} \left(p_\theta(x) \| p_\theta(x;\xi)\right)\right] = \underset{\xi}{\mathbb{E}} \left[\sum_{c\in C} p_\theta^c(x) \log \frac{p_\theta^c(x)}{p_\theta^c(x;\xi)}\right] \tag{7}$$

$$= \sum_{c\in C} \underset{\xi}{\mathbb{E}} \left[p_\theta^c(x) \log \frac{\exp f_\theta^c(x) \sum_{c'\in C} \exp f_\theta^{c'}(x;\xi)}{\exp f_\theta^c(x;\xi) \sum_{c'\in C} \exp f_\theta^{c'}(x)}\right] \tag{8}$$

$$= \sum_{c\in C} \underset{\xi}{\mathbb{E}} \left[p_\theta^c(x)(f_\theta^c(x) - f_\theta^c(x;\xi)) + p_\theta^c(x)(A(f_\theta(x;\xi)) - A(f_\theta(x)))\right] \tag{9}$$

$$= \underset{\xi}{\mathbb{E}} \left[\sum_{c\in C} p_\theta^c(x)(A(f_\theta(x;\xi)) - A(f_\theta(x)))\right] = \underset{\xi}{\mathbb{E}} \left[A(f_\theta(x;\xi)) - A(f_\theta(x))\right] \tag{10}$$

which brings us to the regularizer in Eq. 5.

### 4.3 PEA regularization for semi-supervised learning

PEA regularization works as-is in a semi-supervised setting, as the penalties $\mathcal{V}_i$ do not require label information. We train networks for semi-supervised learning in two ways, both of which apply the objective in Eq. 1 on labeled examples and PEA regularization on the unlabeled examples. The first way applies a $\mathrm{tanh}$-variance penalty $\mathcal{V}^t$ and the second way applies a $\mathrm{xent}$-variance penalty $\mathcal{V}^x$, which we define as follows:

$$\mathcal{V}^t(\bar{y}, \tilde{y}) = ||\tanh(\bar{y}) - \tanh(\tilde{y})||_2^2, \quad \mathcal{V}^x(\bar{y}, \tilde{y}) = \mathrm{xent}(\mathrm{softmax}(\bar{y}), \mathrm{softmax}(\tilde{y})), \qquad (11)$$

where $\bar{y}$ and $\tilde{y}$ represent the outputs of a pair of independently sampled child models, and $\tanh$ operates element-wise. The $\mathrm{xent}$-variance penalty can be further expanded as:

$$\mathcal{V}^x(\bar{y}, \tilde{y}) = \mathcal{D}_{KL}(\mathrm{softmax}(\bar{y})|| \mathrm{softmax}(\tilde{y})) + \mathrm{ent}(\mathrm{softmax}(\bar{y})), \qquad (12)$$

where $\mathrm{ent}(\cdot)$ denotes the entropy. Thus, $\mathcal{V}^x$ combines the KL-divergence penalty with an entropy penalty, which has been shown to perform well in a semi-supervised setting [7, 14]. Recall that at non-output layers we regularize with the "direction" penalty $\mathcal{V}^c$. Before the masking noise, we also apply zero-mean Gaussian noise to the input and to the biases of all nodes. In the experiments, we chose between the two output-layer penalties $\mathcal{V}^t/\mathcal{V}^x$ based on observed performance.

## 5  Testing PEA regularization

We tested PEA regularization in three scenarios: supervised learning on MNIST digits, semi-supervised learning on MNIST digits, and semi-supervised transfer learning on a dataset from the NIPS 2011 Workshop on Challenges in Learning Hierarchical Models [13]. Full implementations of our methods, written with THEANO [3], and scripts/instructions for reproducing all of the results in this section are available online at: http://github.com/Philip-Bachman/Pseudo-Ensembles.

### 5.1  Fully-supervised MNIST

The MNIST dataset comprises 60k 28x28 grayscale hand-written digit images for training and 10k images for testing. For the supervised tests we used SGD hyperparameters roughly following those in [9]. We trained networks with two hidden layers of 800 nodes each, using rectified-linear activations and an $\ell_2$-norm constraint of 3.5 on incoming weights for each node. For both standard dropout (SDE) and PEA, we used $\mathrm{softmax} \rightarrow \mathrm{xent}$ loss at the output layer. We initialized hidden layer biases to 0.1, output layer biases to 0, and inter-layer weights to zero-mean Gaussian noise with $\sigma = 0.01$. We trained all networks for 1000 epochs with no early-stopping (i.e. performance was measured for the final network state).

SDE obtained 1.05% error averaged over five random initializations. Using PEA penalty $\mathcal{V}^k$ at the output layer and computing classification loss/gradient only for the unperturbed parent network, we obtained 1.08% averaged error. The $\xi$-perturbation involved node masking but not bias noise. Thus, training the same network as used for dropout while ignoring the effects of masking noise on the classification loss, but encouraging the network to be robust to masking noise (as measured by $\mathcal{V}^k$), matched the performance of dropout. This result supports the equivalence between dropout and this particular form of PEA regularization, which we derived in Section 4.2.

### 5.2  Semi-supervised MNIST

We tested semi-supervised learning on MNIST following the protocol described in [23]. These tests split MNIST's 60k training samples into labeled/unlabeled subsets, with the labeled sets containing $n_l \in \{100, 600, 1000, 3000\}$ samples. For labeled sets of size 600, 1000, and 3000, the full training data was randomly split 10 times into labeled/unlabeled sets and results were averaged over the splits. For labeled sets of size 100, we averaged over 50 random splits. The labeled sets had the same number of examples for each class. We tested PEA regularization with and without denoising autoencoder pre-training [20][4]. Pre-trained networks were always PEA-regularized with penalty $\mathcal{V}^x$

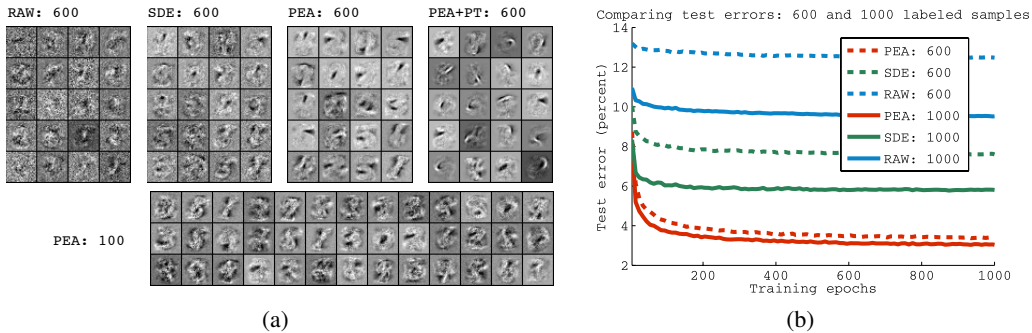

(a)                                                                 (b)

Figure 2: Performance of PEA regularization for semi-supervised learning using the MNIST dataset. The top row of filter blocks in (a) were the result of training a fixed network architecture on 600 labeled samples using: weight norm constraints only (RAW), standard dropout (SDE), standard dropout with PEA regularization on unlabeled data (PEA), and PEA preceded by pre-training as a denoising autoencoder [20] (PEA+PT). The bottom filter block in (a) was the result of training with PEA on 100 labeled samples. (b) shows test error over the course of training for RAW/SDE/PEA, averaged over 10 random training sets of size 600/1000.

on the output layer and $\mathcal{V}^c$ on the hidden layers. Non-pre-trained networks used $\mathcal{V}^t$ on the output layer, except when the labeled set was of size 100, for which $\mathcal{V}^x$ was used. In the latter case, we gradually increased the $\lambda_i$ over the course of training, as suggested by [7]. We generated the pseudo-ensembles for these tests using masking noise and Gaussian input+bias noise with $\sigma = 0.1$. Each network had two hidden layers with 800 nodes. Weight norm constraints and SGD hyperparameters were set as for supervised learning.

Table 1 compares the performance of PEA regularization with previous results. Aside from CNN, all methods in the table are "general", i.e. do not use convolutions or other image-specific techniques to improve performance. The main comparisons of interest are between PEA(+) and other methods for semi-supervised learning with neural networks, i.e. E-NN, MTC+, and PL+. E-NN (EmbedNN from [23]) uses a nearest-neighbors-based graph Laplacian regularizer to make predictions "smooth" with respect to the manifold underlying the data distribution $p_x$. MTC+ (the Manifold Tangent Classifier from [16]) regularizes predictions to be smooth with respect to the data manifold by penalizing gradients in a learned approximation of the tangent space of the data manifold. PL+ (the Pseudo-Label method from [14]) uses the joint-ensemble predictions on unlabeled data as "pseudo-labels", and treats them like "true" labels. The classification losses on true labels and pseudo-labels are balanced by a scaling factor which is carefully modulated over the course of training. PEA regularization (without pre-training) outperforms all previous methods in every setting except 100 labeled samples, where PL+ performs better, but with the benefit of pre-training. By adding pretraining (i.e. PEA+), we achieve a two-fold reduction in error when using only 100 labeled samples.

|       | TSVM  | NN    | CNN   | E-NN  | MTC+  | PL+   | SDE   | SDE+  | PEA    | PEA+   |
|-------|-------|-------|-------|-------|-------|-------|-------|-------|--------|--------|
| 100   | 16.81 | 25.81 | 22.98 | 16.86 | 12.03 | 10.49 | 22.89 | 13.54 | 10.79  | **5.21**   |
| 600   | 6.16  | 11.44 | 7.68  | 5.97  | 5.13  | 4.01  | 7.59  | 5.68  | **2.44**   | 2.87   |
| 1000  | 5.38  | 10.70 | 6.45  | 5.73  | 3.64  | 3.46  | 5.80  | 4.71  | **2.23**   | 2.64   |
| 3000  | 3.45  | 6.04  | 3.35  | 3.59  | 2.57  | 2.69  | 3.60  | 3.00  | **1.91**   | 2.30   |

Table 1: Performance of semi-supervised learning methods on MNIST with varying numbers of labeled samples. From left-to-right the methods are Transductive SVM , neural net, convolutional neural net, EmbedNN [23], Manifold Tangent Classifier [16], Pseudo-Label [14], standard dropout plus fuzzing [9], dropout plus fuzzing with pre-training, PEA, and PEA with pre-training. Methods with a "+" used contractive or denoising autoencoder pre-training [20]. The testing protocol and the results left of MTC+ were presented in [23]. The MTC+ and PL+ results are from their respective papers and the remaining results are our own. We trained SDE(+) using the same network/SGD hyperparameters as for PEA. The only difference was that the former did not regularize for pseudo-ensemble agreement on the unlabeled examples. We measured performance on the standard 10k test samples for MNIST, and all of the 60k training samples not included in a given labeled training set were made available without labels. The best result for each training size is in **bold**.

## 5.3 Transfer learning challenge (NIPS 2011)

The organizers of the NIPS 2011 Workshop on Challenges in Learning Hierarchical Models [13] proposed a challenge to improve performance on a target domain by using labeled and unlabeled

data from two related source domains. The labeled data source was CIFAR-100 [11], which contains 50k 32x32 color images in 100 classes. The unlabeled data source was a collection of 100k 32x32 color images taken from Tiny Images [11]. The target domain comprised 120 32x32 color images divided unevenly among 10 classes. Neither the classes nor the images in the target domain appeared in either of the source domains. The winner of this challenge used convolutional Spike and Slab Sparse Coding, followed by max pooling and a linear SVM on the pooled features [6]. Labels on the source data were ignored and the source data was used to pre-train a large set of convolutional features. After applying the pre-trained feature extractor to the 120 training images, this method achieved an accuracy of 48.6% on the target domain, the best published result on this dataset.

We applied semi-supervised PEA regularization by first using the CIFAR-100 data to train a deep network comprising three max-pooled convolutional layers followed by a fully-connected hidden layer which fed into a $\mathrm{softmax} \rightarrow \mathrm{xent}$ output layer. Afterwards, we removed the hidden and output layers, replaced them with a pair of fully-connected hidden layers feeding into an $\ell_2$-hinge-loss output layer[5], and then trained the non-convolutional part of the network on the 120 training images from the target domain. For this final training phase, which involved three layers, we tried standard dropout and dropout with PEA regularization on the source data. Standard dropout achieved 55.5% accuracy, which improved to 57.4% when we added PEA regularization on the source data. While most of the improvement over the previous state-of-the-art (i.e. 48.6%) was due to dropout and an improved training strategy (i.e. supervised pre-training vs. unsupervised pre-training), controlling the feature activity and output distributions of the pseudo-ensemble on unlabeled data allowed significant further improvement.

## 6   Improved sentiment analysis using pseudo-ensembles

We now show how the Recursive Neural Tensor Network (RNTN) from [19] can be adapted using pseudo-ensembles, and evaluate it on the Stanford Sentiment Treebank (STB) task. The STB task involves predicting the sentiment of short phrases extracted from movie reviews on RottenTomatoes.com. Ground-truth labels for the phrases, and the "sub-phrases" produced by processing them with a standard parser, were generated using Amazon Mechanical Turk. In addition to pseudo-ensembles, we used a more "compact" bilinear form in the function $f : \mathbb{R}^n \times \mathbb{R}^n \rightarrow \mathbb{R}^n$ that the RNTN applies recursively as shown in Figure 3. The computation for the $i^{th}$ dimension of the original $f$ (for $v_i \in \mathbb{R}^{n \times 1}$) is:

$$f_i(v_1, v_2) = \tanh([v_1; v_2]^\top T_i [v_1; v_2] + M_i[v_1; v_2; 1]), \quad \text{whereas we use:}$$

$$f_i(v_1, v_2) = \tanh(v_1^\top T_i v_2 + M_i[v_1; v_2; 1]),$$

in which $T_i$ indicates a matrix slice of tensor $T$ and $M_i$ indicates a vector row of matrix $M$. In the original RNTN, $T$ is $2n \times 2n \times n$ and in ours it is $n \times n \times n$. The other parameters in the RNTNs are a transform matrix $M \in \mathbb{R}^{n \times 2n+1}$ and a classification matrix $C \in \mathbb{R}^{c \times n+1}$; each RNTN outputs $c$ class probabilities for vector $v$ using $\mathrm{softmax}(C[v; 1])$. A ";" indicates vertical vector stacking.

We initialized the model with pre-trained word vectors. The pre-training used $\mathtt{word2vec}$ on the training and dev set, with three modifications: dropout/fuzzing was applied during pre-training (to match the conditions in the full model), the vector norms were constrained so the pre-trained vectors had standard deviation 0.5, and $\mathtt{tanh}$ was applied during $\mathtt{word2vec}$ (again, to match conditions in the full model). All code required for these experiments is publicly available online.

We generated pseudo-ensembles from a parent RNTN using two types of perturbation: subspace sampling and weight fuzzing. We performed subspace sampling by keeping only $\frac{n}{2}$ randomly sampled latent dimensions out of the $n$ in the parent model when processing a given *phrase tree*. Using the same sampled dimensions for a full phrase tree reduced computation time significantly, as the parameter matrices/tensor could be "sliced" to include only the relevant dimensions[6]. During

training we sampled a new subspace each time a phrase tree was processed and computed test-time outputs for each phrase tree by averaging over 50 randomly sampled subspaces. We performed weight fuzzing during training by perturbing parameters with zero-mean Gaussian noise before processing each phrase tree and then applying gradients w.r.t. the perturbed parameters to the unperturbed parameters. We did not fuzz during testing. Weight fuzzing has an interesting interpretation as an implicit convolution of the objective function (defined w.r.t. the model parameters) with an isotropic Gaussian distribution. In the case of recursive/recurrent neural networks this may prove quite useful, as convolving the objective with a Gaussian reduces its curvature, thereby mitigating some problems stemming from ill-conditioned Hessians [15]. For further description of the model and training/testing process, see the supplementary material and the code from `http://github.com/Philip-Bachman/Pseudo-Ensembles`.

|  | RNTN | PV | DCNN | CTN | CTN+F | CTN+S | CTN+F+S |
|---|---|---|---|---|---|---|---|
| Fine-grained | 45.7 | 48.7 | 48.5 | 43.1 | 46.1 | 47.5 | 48.4 |
| Binary | 85.4 | 87.8 | 86.8 | 83.4 | 85.3 | 87.8 | 88.9 |

Table 2: Fine-grained and binary root-level prediction performance for the Stanford Sentiment Treebank task. RNTN is the original "full" model presented in [19]. CTN is our "compact" tensor network model. +F/S indicates augmenting our base model with weight fuzzing/subspace sampling. PV is the Paragraph Vector model in [12] and DCNN is the Dynamic Convolutional Neural Network model in [10].

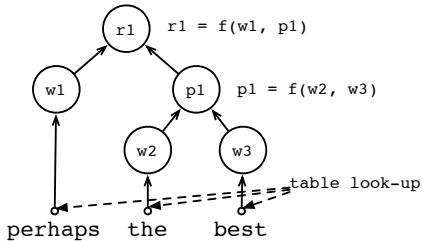

Figure 3: How to feedforward through the Recursive Neural Tensor Network. First, the tree structure is generated by parsing the input sentence. Then, the vector for each node is computed by look-up at the leaves (i.e. words/tokens) and by a tensor-based transform of the node's children's vectors otherwise.

Following the protocol suggested by [19], we measured root-level (i.e. whole-phrase) prediction accuracy on two tasks: fine-grained sentiment prediction and binary sentiment prediction. The fine-grained task involves predicting classes from 1-5, with 1 indicating strongly negative sentiment and 5 indicating strongly positive sentiment. The binary task is similar, but ignores "neutral" phrases (those in class 3) and considers only whether a phrase is generally negative (classes 1/2) or positive (classes 4/5). Table 2 shows the performance of our compact RNTN in four forms that include none, one, or both of subspace sampling and weight fuzzing. Using only $\ell_2$ regularization on its parameters, our compact RNTN approached the performance of the full RNTN, roughly matching the performance of the second best method tested in [19]. Adding weight fuzzing improved performance past that of the full RNTN. Adding subspace sampling improved performance further and adding both noise types pushed our RNTN well past the full RNTN, resulting in state-of-the-art performance on the binary task.

## 7 Discussion

We proposed the notion of a pseudo-ensemble, which captures methods such as dropout [9] and feature noising in linear models [5, 21] that have recently drawn significant attention. Using the conceptual framework provided by pseudo-ensembles, we developed and applied a regularizer that performs well empirically and provides insight into the mechanisms behind dropout's success. We also showed how pseudo-ensembles can be used to improve the performance of an already powerful model on a competitive real-world sentiment analysis benchmark. We anticipate that this idea, which unifies several rapidly evolving lines of research, can be used to develop several other novel and successful algorithms, especially for semi-supervised learning.

## Footnotes

[1]It is easy to formulate analogous objectives for unsupervised learning, maximum likelihood, etc.

[2]Note that "parameters" in a linear program are analogous to inputs in standard machine learning terminology, as they are observed quantities (rather than quantities optimized over).

[3] While dropout is well-supported empirically, its mode-of-action is not well-understood outside the limited context of linear models.

[4]See our code for a perfectly complete description of our pre-training.

[5]We found that $\ell_2$-hinge-loss performed better than $\mathrm{softmax} \rightarrow \mathrm{xent}$ in this setting. Switching to $\mathrm{softmax} \rightarrow \mathrm{xent}$ degrades the dropout and PEA results but does not change their ranking.

[6]This allowed us to train significantly larger models before over-fitting offset increased model capacity. But, training these larger models would have been tedious without the parameter slicing permitted by subspace sampling, as feedforward for the RNTN is $\mathcal{O}(n^3)$.

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
