[Supplementary Material · supplement.pdf]

# Supplemental Material for Learning with Pseudo-Ensembles

**Philip Bachman**
McGill University
Montreal, QC, Canada
phil.bachman@gmail.com

**Ouais Alsharif**
McGill University
Montreal, QC, Canada
ouais.alsharif@gmail.com

**Doina Precup**
McGill University
Montreal, QC, Canada
dprecup@cs.mcgill.ca

## 1    Details for the MNIST tests

See the test code: `http://github.com/Philip-Bachman/Pseudo-Ensembles`.

## 2    Details for the NIPS Transfer Learning Challenge test

Hyperparameters for the convolution/pooling layers were set to maximize supervised performance on CIFAR-100 and hyperparameters for the three layers trained on the target domain were selected via cross-validation on the available training images. Each convolutional layer in our network used 96 5x5 convolutional filters with a stride of 1. Each max pooling layer pooled over 5x5 windows with a stride of 2 and then applied cross-map contrast normalization as in [1]. The fully-connected layer had 1000 nodes. Dropout was applied to max-pooled outputs of the third (final) convolutional layer and the fully-connected layer. All layers used the rectified-linear activation function. We set a max $\ell_2$-norm constraint of 4 on the convolutional filters and the incoming weights for each hidden/output layer node. We trained with SGD using a learning rate of 0.01 and a momentum of 0.9. We trained for 300 epochs, with the learning rate decayed x0.998 after each epoch.

The two fully-connected layers used for supervised training on the target domain each had 200 nodes. Dropout was again applied to the outputs of the third convolutional layer and all nodes in the hidden layers. We applied PEA penalty $\mathcal{V}^t$ on the output layer and $\mathcal{V}^c$ on the hidden layers. The two new hidden layers and the final $\ell_2$-hinge-loss layer were trained by SGD with a learning rate of 0.001 and a momentum of 0.5. We trained for 50 epochs without learning rate decay, then for 25 epochs with the learning rate decayed x0.7 after each epoch. Results for dropout and PEA regularization were averaged over five random initializations.

## 3    Details for the RNTN pseudo-ensemble

The official train/validate/test split provided for the Stanford Sentiment Treebank task, and which we used for our results, is *extremely* unrepresentative of random train/validate/test splits with the same proportions. When randomly re-splitting the dataset 1 million times, the KL-divergence between the sampled train/test set class priors was larger than that of the official train/test split with probability $< 0.00005$. Splits with larger than *half* the KL-divergence of the official split occurred with probability $< 0.01$. When setting hyperparameters using the official train/validate splits, our RNTN pseudo-ensemble actually obtained (slightly) $> 50.0\%$ accuracy on the fine-grained.

# References

[1] Alex Krizhevsky, Ilya Sutskever, and Geoffrey E Hinton. Imagenet classification with deep convolutional neural networks. In *Advances in Neural Information Processing Systems (NIPS)*, 2012.