[Reviews · NeurIPS 2014]

Submitted by Assigned_Reviewer_3

The authors formalize the notion of learning with pseudo-ensemble, under which several existing learning methods can be explained, e.g., dropout in deep neural networks. The authors further present the PEV regularization, which encourages the robustness of learned model (activations of hidden units in different layers in the case of deep layered neural networks) under perturbation of the model space. Since no label information is used in the PEV regularization, it naturally generalizes to semi-supervised learning setup. The authors compare the method to several baseline methods under different learning scenarios, supervised, semi-supervised and transfer learning and show improvement.

The Pseudo-Ensemble notion generalizes previous work on robust learning w.r.t. perturbation in the input space (Burge & Scholkopf 1997, Chapelle et. al. 2000, Maaten et. al. 2013, Wager et. al. 2013). It is a straight-forwarding extension. The PEV regularization is interesting by itself, and is able to match the performance of dropout under supervised learning setup, and significantly outperform in the scenarios of semi-supervised and transfer learning.

The authors try to connect the PEV regularization to dropout in section 4.1. The pseudo-ensemble objective introduced in eq.(1) explains dropout in the limiting case. However, it is less clear from the writings that learning with the PEV regularization actually approximates dropout. It is not very convincing to draw the conclusion that discouraging co-adaption is the reason of success for both PEV regularization and dropout simply by verifying the performance (accuracy numbers) of the two model is similar.

Starting from eq. (1) , a more natural formulation for the regularization would be to penalize the variance of distribution of the output layer rather than summing over all different layers from 2 to d as shown in eq. (3). Could the authors explain why it is done in the current formulation and how much performance deterioration would be observed if we drop the penalties on the hidden layers.

Small comment:
1. Figure 1 is not explained in the text.

2. Based on Table 2, even with the PEV regularization, DAE pre-training (PEV+) was still able to significantly improve the performance of the learned model, which seems to suggest that the PEV regularization is less effective in terms of utilizing unlabeled data.
Summary: The work introduces an interesting regularization for learning with multi-layer neural networks, motivated by dropout and pseudo-ensemble learning. The regularization can also be applied to other classes of models. The authors try to connect directly the PEV regularization to dropout in section 4.1, but the connection is not clear from the writing. Though a straight-forward method, the PEV regularization offers satisfying performance. It could be of interest to a subset of the community to learn about it.

Submitted by Assigned_Reviewer_20

In essence, this paper considers the generalization of dropout to other classes of models by perturbing the parameters of the corresponding source model. For instance, in a the simple case of a mixture model, one could add appropriate noise to the means of the components or to their covariance matrices. The paper introduces a regularizer to ensure robustness of the pseudo-ensemble of child models with respect to the noise process used to perturb the parent model and generate the child models. The general topic of the paper is of significant interest to the NIPS community and in my opinion worth presenting, although overall it is a somewhat straightforward generalization of dropout. The ideas are supported by several informative simulations on standard benchmark datasets.

Some specific comments:

1)The first two sentences are too vague to make sense, especially in relation to the role of the variables x and y. For instance, is one trying to derive approximations of p(x,y) that are conditioned on x?

2) The statement "Many useful methods could be developed....by generating perturbations beyond the iid masking noise that has been considered for neural networks" is not correct. Even in the original paper by the Hinton group the noise was not iid since the dropout probability used in the input layer (data) was lower (e.g. 0.2) than in the other layers (e.g. 0.5). Furthermore, examples of non-iid cases were analyzed in reference [1] which should be augmented with, or replaced by, its more extended version (Artificial Intelligence, 210, 78–122, 2014). This reference considers also other forms of noise, such as adding Gaussian noise to the activity of the units, and shows how they the fall under the same framework. This is highly related to the theme of this paper.

3) This is a minor point but the author may want to consider changing the terminology. The term "pseudo-ensemble" is perhaps not ideal since "pseudo" has a slightly negative connotation, whereas the point to convey is that this is a new approach to learning in its own right.

4) The remark "While dropout is well supported empirically, its mode of action is not well understood outside the limited context of linear models" is not entirely correct. In the non-linear case, the ensemble properties of dropout in deep non-linear neural networks are reasonably well understood, and so are its regularization properties, as described in the reference given above.

5) This is another minor point but there are a few typos. The paper should be run through a spell-checker. See, for instance, the last line of page 6 ("northe images in the target domain").

6) The paper is well supported by several informative experiments on different benchmark datasets.

Summary: In essence this paper presents an incremental generalization of dropout. This is currently a hot topic for the NIPS audience. It is supported by a set of interesting simulations.

Submitted by Assigned_Reviewer_28

This paper proposes the notion of a pseudo-ensemble, that of dependent child models ensembled together, which is an nice and unified way of several related techniques prominently including dropout. Then presented the pseudo ensemble variance regularizer and tested it with convincing empirical results. The presented technique seems much easier to apply to multi-layer, semi-supervised setting than the technique of [23].

Some objections are that the notion of this pseudo-ensemble as an expectation is already in references [5] and [23], so the contribution of naming it and generalizing to multi-layer is not a significant leap. The authors made an effort of gaining more understanding by writing down the boosty and baggy forms, but nothing was really done with them and the connection to boosting is vague. Finally, while it is okay to introduce an regularizer and show that it is empirically successful, it would be nice if it corresponds to, or is approximating something that we understand. The structure of paper would have me believe that this is done, as sections 1-3 talks about a general framework, and section 4 should be testing it. Instead, section 4 introduces a new regularizer that has little to do with the previous sections. I would have given the paper a higher score based on its empirical strength and the simplicity of the proposed regularizer compared to the alternatives, if not for the fancy presentation that made it harder to read but did not add much insights. Please explain in the response if I am wrong on this and I am willing to change my scores.

Detailed comments:
the formalism of f(x; \xi) is much more general than the situation the paper seek to deal with: that of subsampling child models through \xi. Without alluding to that \xi represents something like dropout, this formalism does not mean much. Unfortunately, this seems hard to remedy without making the notations more complicated, so perhaps it is okay.

Line 106 "Note how the left objective moves the loss L inside the expectation over noise"...
this contradicts the equation, where the loss L is outside the expectation over noise

Eq 2): baggy PE does not seem to make sense to me, are you summing over i? or is one of the expectations supposed to be taken over i as well?

While there is some intuitive resemblance (final prediction is made by the sum of weak learners, though the sum is weighted in boosting), it is unclear to me how these boosty PE concretely connect to boosting.
The flavor of sequentially adding more classifiers is not there, nor is the boosting weights assigned to each example. Can you make this more concrete? If not, perhaps it would be good to say this is how you define boosty in the first place and the reader should not be looking for rigorous connections.

In line 129-140, you compared a few existing approaches and say that they only deal with the input space. I do not think this is entirely true. My perspective on this:
A: [5] Learning with MCFs: non-convex (convex in some cases) lower bound by
moving the expectation inside the log but not inside the exp.
B: [23] adaptive regularization: delta method (second-order central limit)
expansion (relies on 4th moments small). This has the advantage of giving an explicit regularizer that's interpretable. Non-convex.
C: Wang and Manning, Fast Dropout: relies on central limit
theorem and sum of noise converging to Gaussian. Non-convex and deals with model noise as well as input noise.
D: Baldi and Sadowski, Understanding dropout paper, analyzes geometric mean, also deals with model noise.

Eq 3): why is there a subscript on the variance notation? it is the same operation regardless of i (correction: not so in line 210: is it worth the subscript though).

You should perhaps call \mathcal{V} scale-invariant variance penalty, so the reader does not think you are talking about the actual variance. The font helps, but you still said its variance.

What is the motivation of this regularization method? can you show that this converges to [23] for linear models somehow? or can you provide more understanding by somehow deriving/approximate it instead of just defining it?
Summary: This paper presented the pseudo ensemble variance regularizer and tested it with convincing empirical results. The PEV regularization technique seems much easier to apply to multi-layer, semi-supervised setting than the technique of [23]. However, the regularizer is unmotivated theoretically and very little is done with the conceptual discussions of boosting and bagging while the presentation would lead readers to believe otherwise.

Submitted by Assigned_Reviewer_45

Summary:
This paper proposes a regularization technique for neural nets where the model
is encouraged to reduce the variance of each hidden layer representation over
dropout noise being added to the layers below. This idea is generalized to
``pseudo-ensemble" models where other kinds of perturbations can be used.

The main contribution of this paper is the variance regularizer. Experiments
are done on MNIST (supervised and semi-supervised) and NIPS'11 transfer
learning dataset (CIFAR-100, TinyImages) using standard neural nets with
dropout perturbations. The authors also experiment with the Stanford Sentiment
Treebank dataset using Recursive Neural Tensor Nets with other kinds of
perturbations. The experiments show that this regularizer works the same or
better than using the perturbations alone.

Strengths-
- The model gets promising results in harsh situations where there is very
little labelled data.
- The experiments are well chosen to highlight the applicability of this method
to different models and datasets.

Weaknesses -
- Some parts of the paper seem somewhat superfluous. It's not clear what the
discussion about Baggy/Boosty PE adds to the paper (assuming that the main
point is the variance regularizer).
- Some crucial details about the experiments should be included. Those are
mentioned below.

The authors should mention / discuss -
(1) How many noise samples were used to compute the different variances ?
(2) Was back-prop done through each dropped-out model or just the clean one ?

(3) One of the major problems of dropout is that it slows down training. This
approach probably further exacerbates this problem by requiring that one must
do multiple forward and (back ?) props per gradient update (with/without noise,
or with different noise samples to compute the variance). It would be good to
analyze how much of a slow-down we get, if any, by making a plot of
training/test error vs time (as opposed to number of epochs).

(4) What was the stopping criterion for the semi-supervised MNIST experiments ?
The previous section mentions "We trained all networks for 1000 epochs with no
early-stopping." Does that also apply to the semi-supervised experiments ? If
yes, was it kept 1000 epochs even for 100 labelled cases ? It seems likely that
500-500 or 1000-1000 nets would overfit massively on the labelled data sizes
considered here, even with dropout, by the end of 1000 epochs for reasonable
learning rates. Is this true ? I think it is very important to know if early
stopping with a validation set was needed because in small data regimes, large
validation sets are hard to find.

(5) If multiple (say $n$) forward and backprops are done per gradient update in
PEV, would it be fair to compare models after running for a fixed number of
epochs ? Wouldn't each PEV update be equivalent to roughly $n$ regular SDE
updates ?

(6) For the semi-supervised case, did each mini-batch contain a mixture of
labelled and unlabelled training cases ? If yes, what fraction were labelled ?

(7) Consider comparing with SDE+ in Table 1 ?

(8) Was the same architecture and dropout rate used for SDE and PEV in Table 1
? If yes, is that a fair comparison ? May be it's possible that for SDE, a
smaller net or the same net with higher dropout would work better ? It is clear
that PEV is a ``stronger" regularizer, so we should probably also let SDE be
strong in its own way (by having higher dropout rate).

Quality:
The experiments are well-designed. Some more explanations and comparisons, as
asked for above, will add to the quality.

Clarity:
The paper is well-written barring minor typos.

Originality:
The variance regularizer is a novel contribution.

Significance:
This paper could have a strong impact on people working with small datasets.
Summary: This paper proposes an interesting way of regularizing models. The experiments are convincing, but the paper can be improved by adding some more details and clarifications.
Author Feedback
Author rebuttal: Thank you for the helpful comments.

To Reviewer 20:
We've corrected typos and grammatical errors.
1. We re-worded this, e.g. the first sentence of Sec.2 changed from "Consider a data distribution p_{xy} which we want to approximate using a parametric parent model f_{theta}" to: "Consider a function f(x), e.g. a distribution p(x) or conditional distribution p(y|x), which we want to approximate using a parametric parent model f_{theta}."

2. Yes, we should say "indedependent noise", rather than "iid" noise, as drop rates in deep networks may differ between layers. We've had time to look over the extended version of [1] and will account for it.

3. We agree and will find a better name.

4. Yes, the extended version of [1] looks at effect of dropout in deep non-linear networks as adaptive L2 regularization.

To reviewer 28:
Various instances of pseudo-ensembles (PEs) exist in the literature. We find it's helpful to unite related concepts under a named category to facilitate a refactoring of our conceptual toolboxes (see "Self-taught learning" as an example). Then, we can more clearly see what methods in the category have in common, and how the category is distinct from related ones. Along these lines, we will work to improve the boosty/baggy pseudo-ensemble part, in clarity of both presentation and purpose.

Our goal in presenting the boosty pseudo-ensemble was to highlight one key difference between PEs and boosting: that boosting only considers the expected ensemble output and doesn't offer an explicit mechanism for controlling other aspects of ensemble behavior. The notion of explicitly shaping distributional properties of an ensemble is easily motivated in the PE framework, but much less obvious via boosting. The way in which PEV regularization becomes natural after assimilating the idea of a PE also links the "conceptual" and "empirical" portions of our paper. We will make it clear that we aren't addressing the stagewise optimization or input reweighting aspects of boosting.

For line 106: yes, we've corrected the wording here.

For the baggy PE, the superscript i indicates that each sample from the noise process is associated with its own parameters and input distribution, e.g. a bootstrap sample from the training distribution. Estimation for each i is independent and the ensemble output is computed in expectation with respect to the xi^i.

We may rename PEV to avoid confusion between our colloquial use of "variance" and the precise statistical notion.

We have a derivation showing that regularizing the KL-divergence of a PE's predictions with respect to the parent model's predictions is equivalent to standard dropout when both methods assume softmax + cross-entropy at the output layer. This holds as long as the PE's expected pre-softmax output is equal to that of the parent model. This formally relates PEV to dropout and the work in [23]. We will include this derivation in our paper.

To Reviewer 3:
The equivalence between PEV regularization and dropout in some settings will be formalized in our final draft (see comments above). We would also like to formalize a reasonable measure of feature "co-dependence" that is leveraged by PEV regularization, but this goes beyond the scope of the current paper.

We include the possibility of regularizing activity over all layers of a deep network for generality. We think of activations in the hidden layers as latent variables/features and it seems reasonable to control their distributional properties in addition to those of the model's class predictions. We think this helps to show how PEV regularization might be applied in models (e.g. generative graphical models mixing latent/observed variables) that don't have a strong separation between input/output/latent variables.

We will describe Figure 1 in the text of our final draft and add SDE+ to the semi-supervised MNIST results.

To reviewer 45:
Please see the replies to the other reviewers, which also address your conceptual points.

1. For each input, we compute the PEV gradient by comparing a single pass through the noisy model to a single pass through the noiseless model.
2. Each feedforward pass gets its own backprop pass.
3. There's a tradeoff. Empirically, in the fully-supervised setting the PEV-regularized model converges more quickly than dropout in terms of epochs/updates, but not much differently in terms of clock time. This might be because PEV passes classification loss/gradients through the noise-free network. In the s-s setting PEV takes longer to converge to its eventual result, but the result is significantly better for PEV.
4. We didn't use early stopping in any of our experiments. We trained for 1000 epochs in all settings except those where "staged" training was used (e.g. pretrain + 100 labeled samples). During "staged" training 10+10+10+500 epochs were used for s-s learning. Performance was measured based on the final model. Note: we use "epoch" to mean 250 minibatch updates (to avoid confusion when the labeled training set changes size).
5. A PEV update has ~2x the cost of a dropout update.
6. In s-s learning each minibatch was half labeled and half unlabeled. Each labeled input only passed through the noisy network, and each unlabeled input passed once through the noisy and noiseless networks.
7. See reply to Reviewer 3.
8. The architecture and training hyperparameters (e.g. learning rate, momentum, etc.) were selected using standard dropout. Only the staged training involved specific choices optimized for PEV performance. For completeness, we can add some experiments in the supplement examining different architectures.